# Beige Fat, Adaptive Thermogenesis, and Its Regulation by Exercise and Thyroid Hormone

**DOI:** 10.3390/biology8030057

**Published:** 2019-07-31

**Authors:** Kevin J. Phillips

**Affiliations:** Department of Molecular and Cellular Biology, Baylor College of Medicine, Houston, TX 77030, USA; kevinp@bcm.edu

**Keywords:** beige fat, brown fat, thermogenesis, adipose, exercise, thyroid hormone

## Abstract

While it is now understood that the proper expansion of adipose tissue is critically important for metabolic homeostasis, it is also appreciated that adipose tissues perform far more functions than simply maintaining energy balance. Adipose tissue performs endocrine functions, secreting hormones or adipokines that affect the regulation of extra-adipose tissues, and, under certain conditions, can also be major contributors to energy expenditure and the systemic metabolic rate via the activation of thermogenesis. Adipose thermogenesis takes place in brown and beige adipocytes. While brown adipocytes have been relatively well studied, the study of beige adipocytes has only recently become an area of considerable exploration. Numerous suggestions have been made that beige adipocytes can elicit beneficial metabolic effects on body weight, insulin sensitivity, and lipid levels. However, the potential impact of beige adipocyte thermogenesis on systemic metabolism is not yet clear and an understanding of beige adipocyte development and regulation is also limited. This review will highlight our current understanding of beige adipocytes and select factors that have been reported to elicit the development and activation of thermogenesis in beige cells, with a focus on factors that may represent a link between exercise and ‘beiging’, as well as the role that thyroid hormone signaling plays in beige adipocyte regulation.

## 1. Introduction

Obesity and the co-morbidities of metabolic syndrome are an accelerating worldwide health crisis. Since the withdrawal of the popular diet cocktail fen-phen (a mixture of fenfluramine and phentermine) few weight loss strategies aside from surgical intervention have come close to matching fen-phen’s efficacy [1]. Perhaps all weight loss therapeutics to date have altered energy balance by decreasing caloric intake. There has long been an allure in utilizing thermogenesis, the conversion of energy (such as excess caloric intake) to heat, as a weight loss strategy that functions by increasing energy expenditure. *Can thermogenesis be used as a therapeutic anti-obesity strategy?* The answer to this question is not fully clear. Increasing energy expenditure via thermogenesis brings with it the obvious risk that the additional heat that accompanies an increased metabolic rate could result in hyperthermia, an acute risk not shared with strategies that decrease caloric intake. This risk is borne out by the misuse of the chemical uncoupler dinitrophenol, which has been shown to be highly efficacious in producing weight loss, but has also resulted in overdose deaths [2,3,4]. Thus, it may be challenging for any thermogenic-based therapeutic to clear necessary safety and regulatory hurdles. One suggestion to the challenge of safely inducing thermogenesis has been to utilize the increased heat loss caused by processes such as vasodilation to trigger thermogenesis. This perspective highlights an oft-overlooked component of thermogenesis, that physiological actions that affect heat loss and retention will generally accompany alterations in the systemic metabolic rate.

Weight loss is known to elicit reductions in the metabolic rate [5,6] and a recent study, featuring contestants of the game show “Biggest Loser”, reported that weight loss can elicit metabolic adaptations that includes large and long lasting reductions in metabolic rate [7]. In instances such as these, where metabolic rate is remarkably low, thermogenic agents that increase basal metabolic rate may be desirable, if not necessary, to sustain weight loss. Technically, one strategy to induce thermogenesis is already well known—cold exposure. As body temperature is maintained at a fairly fixed temperature, controlled by the hypothalamus, endothermic species placed under significant cold stress are required to increase metabolic/heat output to maintain defended body temperature. This increased heat output is termed cold-induced thermogenesis and consists of two major contributors, shivering and non-shivering or adaptive thermogenesis. Given the unpleasant nature of shivering, it is not likely a viable weight loss therapy. Thus, this review will focus on adaptive thermogenesis [8]. Adaptive thermogenesis is mediately principally (if not wholly) by the protein uncoupling protein 1 (UCP1). When activated, UCP1 uncouples mitochondrial respiration from ATP generation. Thus, the potential energy generated by respiration is no longer used to conduct the work of ATP production and is instead lost as heat. Whether non-UCP1-mediated mechanisms of adaptive thermogenesis exist is a matter of debate and is discussed below. UCP1 expression is limited to brown adipose tissue (BAT) and beige adipocytes, which reside within white adipose tissue (WAT).

## 2. Adipose Tissues

### 2.1. White Adipose Tissue

The principal function of WAT, energy homeostasis, requires the performance of two opposing yet related actions, the storage of energy as triglycerides in times of excess caloric intake and the release of energy for use by other tissues in times of caloric deficit. In times of energy deficit, such as during fasting, white adipocytes break down triglycerides into fatty acids and glycerol, which are released into circulation and used by other tissues [9]. Thus, WAT function is crucial for proper energy balance and lipid homeostasis. Key regulators of adipocyte lipolysis are glucagon and the sympathetic nervous system (SNS), which releases the β-adrenergic receptor (β-AR) agonist norepinephrine (NE), both of which stimulate the lipolysis of triglycerides. Agonism of β-ARs initiates a signaling cascade involving the activation of protein kinase A (PKA), which ultimately results in the breakdown of triglycerides. Conversely, in times of caloric excess, insulin levels increase, and insulin becomes the predominant regulatory factor. Insulin stimulates the opposing response, preventing lipolysis and promoting lipogenesis, the synthesis and storage of triglycerides

In addition to providing a storage depot for energy, WAT also serves as an endocrine tissue, secreting hormones, the most well-known being leptin [10]. The ability of WAT to serve as an energy sink in times of caloric excess, serves an important secondary purpose, protecting non-adipose tissues, notably liver and muscle, from lipid burden, which is deleterious and often termed lipotoxicity. Thus, healthy adipose tissue expansion is beneficial and can prevent maladies such as hepatosteatosis or fatty liver, which contributes to insulin resistance and diabetes. WAT expansion occurs via two mechanisms, hyperplasia, the production of new adipose cells via de novo lipogenesis, and hypertrophy, the growth of existing cells. Evidence suggests that different WAT depots expand via different mechanisms [11,12]. Mice fed an high fat diet (HFD) responded to the caloric excess via a combination of both hyperplasia and hypertrophy of visceral adipocytes, while subcutaneous depots did not display hyperplasia, and instead, grew solely via hypertrophy of preexisting adipose cells. Thus, WAT depots can have a diverse responses to similar growth stimuli. Regardless of the mechanism, adipose expansion cannot proceed indefinitely, and prolonged periods of caloric excess will overrun WATs’ ability to store the excess energy, leading to adipose dysfunction, potential adipocyte death, inflammation, and fibrosis within adipose tissues. When this occurs, the excess energy that cannot be stored in adipose tissues is diverted to extra-adipose tissues, resulting in the aforementioned lipotoxicity, with this overall phenomenon being a root cause of metabolic syndrome.

### 2.2. Brown Adipose Tissue

In addition to WAT, nearly all placental mammals also possess thermogenically competent BAT [13], which as the name suggests, has a physical and morphological appearance quite distinct from WAT [14]. This difference represents the divergent function of BAT. While WAT serves as a tissue for energy storage, BAT, which is abundant in hibernating animals, serves to burn excess energy and generate heat. Large BAT depots are present in mice and other rodents and are used to maintain body temperature during periods of cold stress. In contrast to white adipocytes, which have few mitochondria and a relatively low metabolic rate, BAT contains a high abundance of mitochondria and is highly vascularized [15], giving it its brown color and allowing the tissue to maintain a high level of respiration. The hallmark of BAT is that it is the only tissue, under normal conditions, to contain the protein uncoupling protein 1 (UCP1) [16,17]. UCP1, when activated, allows brown adipocytes to conduct uncoupled respiration, the ‘uncoupling’ of respiration and electron transport chain activity from the generation of ATP. Thus, UCP1 creates a futile cycle where respiration no longer performs its usual work of producing ATP and instead discards of the energy produced from respiration as heat, a process referred to as non-shivering thermogenesis. Since this process of uncoupled respiration disrupts the ability of cells to produce their normal cellular energy currency of ATP, it should be obvious that the process must be carefully regulated, which also provides a rationale for why UCP1 expression is highly restricted to expression only in adipocytes. 

The primary and most well studied mechanism of BAT regulation is via the SNS. During periods of cold stress, nerves of the SNS produce norepinephrine at sites of innervation within BAT, which binds to the β-ARs on brown adipocytes and activates BAT. This SNS/β-AR stimulation results in the expression of genes of the thermogenesis program, stimulates BAT expansion, and activates lipolysis of triglycerides within the brown adipocytes. The fatty acids generated from lipolysis, in turn, serve as both activators of UCP1 and as fuel for respiration in brown adipocytes. It used to be thought that BAT relied primarily on the catabolism of intracellular fatty acids as a fuel substrate. However, it is now apparent that brown adipocytes are metabolically flexible as to the energy source utilized [18,19]. Brown adipocytes utilize both glucose and lipids to fuel thermogenesis, the former being routinely utilized for in vivo imaging of BAT activity, and recent studies have suggested that intracellular lipolysis is not required for BAT activity. Genetic deletion of ATGL, a key triglyceride lipase in brown adipocytes that was thought to be necessary for thermogenesis, has been shown to be dispensable for BAT thermogenesis [20,21]. In addition to fatty acids generated intracellularly, brown adipocytes can obtain fatty acids from circulation, which then support thermogenesis. Another study demonstrated that acyl carnitines, produced by the liver, can also be uptaken by BAT and used as a substrate for thermogenesis [22].

Given the ability of uncoupled respiration mediated by BAT and UCP1 to dispense of excess caloric intake as heat, there has long been a desire to harness this effect to oppose obesity and metabolic syndrome. Indeed, studies in mice vie for the validity of this approach. In mice, the loss of UCP1 leads to obesity [23,24], while BAT activation improves insulin sensitivity and serum lipid profiles, and is anti-obesogenic. The sparse data available suggests that BAT activity and these beneficial effects are also relevant in humans [25,26]. It is now accepted that BAT exists in humans, however, the amount of BAT is low, thus leading to questions regarding the extent of the systemic metabolic effects that could be achieved in humans. Thus, it is not clear how effective the activation of human BAT would be in countering obesity. Current enthusiasm may be more focused towards the anti-diabetic effects of BAT, which can lower blood glucose levels when activated.

### 2.3. Beige Adipocytes

In addition to white and brown adipocytes, under certain conditions ‘beige’ adipocytes can also be observed. Beige cells are brown-like adipocytes that appear within WAT depots, a phenomenon referred to as browning or beiging [27,28]. Beiging is induced via similar stimuli as those that lead to BAT recruitment, namely cold and SNS/β-AR agonism, which are the most effective and well appreciated inducers of beige fat. These stimuli lead to the emergence of beige cells in WAT, which, aside from their distinct locale, demonstrate most characteristic features of brown adipocytes. Beige cells have a morphology similar to that of brown adipocytes, with multilocular lipid droplets and a mitochondrial abundance that is more similar to that of brown adipocytes than white adipocytes. Expression of genes of the thermogenic program, most notably Ucp1, is also strongly induced in beige adipocytes relative to white adipocytes, with some reports suggesting that UCP1 protein levels can approach that found in classical brown adipocytes [29]. One major distinction of beige adipocytes has been that they arise from the same Myf5- precursors as white adipocytes. More recently, it has been observed that there is heterogeneity in their origin dependent on adipose depot, with beige adipocytes from the anterior-subcutaneous, retroperitoneal, and posterior-subcutaneous WAT arising from Myf5+ precursors [28]. Beiging was first reported over 30 years ago based on the morphological appearance of brown-like cells in WAT of BALB/c mice [30]. It is now known that the magnitude of beiging is both WAT depot and strain dependent, with inguinal WAT being the most readily beiged in oft-used C57BL/6 mice [31,32]. Recent reports indicate that beige adipocytes, when fully stimulated, can functionally mimic the metabolic actions of classical brown adipocytes [29]. However, the ultimate importance of beige adipocytes in mediating overall systematic metabolism is currently much debated [33].

## 3. Select Beiging Agents

### 3.1. Exercise and Environmental Stimulation

Several reports have suggested that exercise [34,35,36] and environment stimulation [37] can induce the browning of WAT. Especially in the case of exercise, a rationale for this effect is not obvious, since exercise imposes an energy demand upon the organism, it would seemingly be counterintuitive to concurrently induce adaptive thermogenesis, another energy utilizing and heat producing process. Apparent beiging may be due to the fact that the gene expression profile required to elicit changes to WAT required during periods of high energy demand, such as exercise, are similar to those needed for UCP1-dependent thermogenesis, such as an increased requirement for triglyceride lipolysis and changes in lipid handling as well as architecture. Thus, the same gene expression program may be utilized in response to both demands. Consistent with this idea, multiple reports have indicated that morphological and expression changes consistent with beiging occur in WAT of UCP1KO mice following cold exposure [38,39,40]. Stanford et al. reported that exercise, in the form of voluntary wheel running, for 11 days resulted in a substantial beiging in WAT [35]. Analysis of subcutaneous WAT of trained mice revealed changes in gene expression and tissue morphology consistent with beiging as well as increased oxygen consumption in the tissue ex vivo. They showed further that transplantation of this tissue into sedentary mice produced dramatic improvements in glycemic control, while transplants from unexercised mice had no effect. Similarly, Bostrom et al., in reporting the discovery of irisin (discussed below), showed that exercise, either in the form of wheel running or swimming, resulted in large, 25–65 fold, increases in Ucp1 levels in inguinal WAT of trained mice [36]. In 2011, Cao et al. reported that providing a more stimulating environment, furnished with a running wheel, toys and mazes, led to wat beiging and substantial fat loss [37]. This effect was not solely due to wheel running, as mice provided only a running wheel ran much further than mice in the enriched environment, yet showed lesser effects than the environmentally stimulated mice. The beiging was substantial and even more pronounced than that elicited by cold exposure. The effects of environmental stimulation were blocked by propranolol, strongly suggesting that the effects are SNS mediated. The authors concluded that BDNF signaling in the hypothalamus resulting from the environmental stimuli led to the SNS modulation of WAT.

### 3.2. FGF21

Of all the reported beiging agents, FGF21 is perhaps the most well studied and may provide the strongest link between exercise and WAT browning, as numerous reports have indicated that FGF21 levels are increased by exercise [41,42,43,44]. FGF21 analogues have progressed to the point of being investigated in multiple human trials for the indications of obesity, diabetes, and other metabolic disorders [45,46,47,48]. The metabolic actions of FGF21 were first reported in 2005 by Kharitonenkov et al., who showed that pharmacological administration could ameliorate hyperglycemia and hypertriglyceridemia in obese mice [49]. In addition to acting as an autocrine and paracrine factor, like most FGFs, FGF21 can also act as a secreted, circulating hormone [50]. Hepatic production of FGF21 is the predominant source of circulating FGF21, although it is also produced in the pancreas, BAT, and WAT, tissues where it appears to exert autocrine and paracrine actions [51]. FGF21 has been shown to have numerous beneficial effects in ameliorating metabolic disorders. It has been shown to elicit weight loss, increase insulin sensitivity [49,52,53], and reduce serum triglyceride and cholesterol levels in both mice and monkeys [48,54,55,56]. Anti-obesity and anti-diabetic effects have been reported in some, but not all, human trials of FGF21 analogues [46,48]. All studies, however, demonstrated lipid lowering [45,47].

In mice, it is clear that FGF21 induces thermogenesis in BAT and beige adipocytes [52,53,57,58] and stimulates glucose uptake in BAT [53,59]. FGF21’s thermogenic effects on adipocytes appear to be cell-autonomous, as they have been recapitulated in vitro [49,60,61,62]. Accordingly, adipose specific knockout of β-Klotho, a component of FGF21’s receptor, abrogated FGF21s acute effects on insulin sensitivity [59,63,64]. However, adipose specific β-Klotho knockout did not prevent the ability of FGF21 to mediate weight loss, indicating that FGF21 has effects outside of adipose tissues, namely the brain. FGF21’s central effects appear to be mediated by increased SNS output. Intracerebroventricular (ICV) injection of FGF21 resulted in increased NE turnover in WAT and BAT and led to WAT beiging [57,65]. These effects could be blocked in obese mice via the administration of a β-AR antagonist and beiging was lost in β-less mice that lack all β-ARs. Thus, it seems clear that FGF21’s effects on the metabolic rate, at least in mice, are largely mediated by increased SNS output, explaining the observed effects on BAT and WAT. Several studies tested the necessity of adipose thermogenesis for the metabolic effects of FGF21. Kwon et al. reported that FGF21-stimulated glucose clearance was lost in UCP1KO mice [66]. Samms et al. revealed that FGF21-stimulated increase in the metabolic rate was lost in UCP1KO mice [67]. Contrastingly, Venient et al. reported that FGF21’s ability to increase energy expenditure was maintained in the same mice [56]. Collectively, the data seem to suggest that both UCP1-dependent and UCP1-independent effects play a role in the overall metabolic actions of FGF21.

### 3.3. Irisin

As discussed, exercise has been reported to elicit changes to WAT that are consistent with beiging. These include reduction in adipocyte size, mitochondrial biogenesis, and changes in gene and adipokine expression profile, including the induction of UCP1 [68]. In 2012, Bostrom et al. described the discovery of irisin, reported to be an exercise-induced myokine secreted from muscle in response to exercise [36]. Irisin is a proteolytic cleavage product of fibronectin type III domain- containing protein 5 (FNDC5), localized on the surface of white adipocytes and skeletal muscle cells. Following stimulation by exercise, FNDC5 is proteolyzed, releasing irisin into circulation. Irisin was reported to be detectable in blood of both humans and mice, with exercise increasing serum levels. This action was reported to be sufficient to induce WAT beiging in mice, while neutralization of irisin with an antibody to FNDC5 blocked the irisin stimulated expression of thermogenic genes. Other groups have reported similar effects of irisin on WAT browning [69,70]. Despite this fact, it should be noted that the function and pharmacologic significance of irisin is generally regarded as highly questionable. Loss-of-function animal models causally linking irisin or the irisin receptor to WAT beiging are still lacking and concerns persist regarding the ability of exercise to increase serum irisin levels as well as the antibody-based methods used to detect irisin [71]. For the time being, irisin may well be what several articles have referred to it as, a “Greek myth” [72,73]. 

### 3.4. IL-6

IL-6 is expressed in muscle, induced upon muscular contraction, and secreted into circulation. Thus, it has long been considered a potential factor linking exercise to beneficial effects in tissues outside of muscle [74], with more recent studies investigating the potential for IL-6 to mediate WAT beiging. Thermogenic stimuli that activate BAT lead to increased IL-6 expression [75] and IL-6 released from BAT into circulation is thought to influence the metabolism of other tissues including WAT, resulting in beiging. This is supported by the demonstration that IL-6KO mice become obese on HFD [76]. IL-6 has also been reported to be released by muscle following exercise, leading to the induction of lipolysis and altered gene expression in WAT and improvements in whole body insulin sensitivity [77]. Conversely, IL-6 overexpression results in decreased fat mass and adipocyte cell size in mice fed either chow or HFD [78]. The transplantation of BAT in mice has been shown to be beneficial to metabolic health. In order to test whether BAT-derived IL-6 played a role in the beneficial effects of transplantation, Stanford et al. transplanted BAT from IL-6KO mice into age-matched control mice [79]. While transplantation of BAT from wild-type controls had beneficial effects on body weight and insulin sensitivity, these effects were lost with BAT transplantation from IL-6KO mice. Thus, the authors concluded that release of IL-6 from the transplanted BAT was necessary for the observed effects. However, complicating interpretation was the finding that production of another BAT related beiging factor, FGF21, was also lost in the IL-6KO transplants. Thus, the effects of IL-6 may be mediated, at least in part, by FGF21. In studies directly assessing WAT beiging by IL-6, Knudsen et al. reported that IL-6KO mice have reduced expression of UCP1 in inguinal WAT depots following cold exposure or exercise [80]. IL-6KO mice have also been reported to have impaired WAT browning following burn injury relative to WT controls [81].

### 3.5. Succinate

Recently, Mills et al. have reported that the administration of the metabolic intermediate succinate can activate both brown and beige fat UCP1-dependent thermogenesis both in cells and in vivo [82]. The study builds upon recent efforts suggesting that intracellular reactive oxygen species (ROS) can act as a regulator of cellular metabolic processes. The study relies on the premise that, as the redox status of a cell changes, the redox potential of cellular thiols is altered, favoring oxidative modification to cysteine and other cellular thiols [83]. The authors begin by using metabolomic profiling to identify that, during periods of cold induced thermogenesis, levels of succinate increase dramatically in BAT. They show further that succinate increases mitochondrial respiration in brown adipocytes and that succinate itself is used as a substrate by brown adipocytes. The ability of succinate to increase BAT activity and induce WAT beiging reveals that succinate can be taken up from the circulation and used by thermogenic adipocytes, an action that is dependent upon succinate dehydrogenase. The same team reported that the thermogenic activation by succinate is mediated at least in part by the ROS driven modification of a specific cysteine residue, C253, on UCP1 [84]. Using contemporary mass-spectrometry-based proteomics methods, they identified UCP1 C253 to be oxidatively modified following thermogenic stimuli. They propose that C253 modification serves as an allosteric rheostat to sensitize UCP1 to fatty acids and potentially other activators such as purine nucleotides. The authors further postulate that similar ROS driven modifications could act as thermogenic effectors by modifying other factors involved in thermogenesis. It should be noted that the premise of thermogenic induction leading to increased adipocyte ROS is not wholly uncontested. Shabalina et al. have reported that redox stress in brown adipocytes is not sufficient to activate UCP1-dependent thermogenesis [85]. Stier et al. investigated how cold-stimulated adaptive thermogenesis affected cellular redox state. In this study, the authors were unable to find indicators of oxidative stress or changes in thiol status in brown adipocytes from UCP1+ mice, concluding that the reduction in membrane potential that results from UCP1 activation offsets any increased oxidative stress elicited from increased respiration rate [86]. 

### 3.6. β-aminoisobutyric Acid 

β-aminoisobutyric acid (BAIBA) was identified using a mass spectrometry-based approach designed to find exercise-induced factors that might affect metabolic changes in adipocytes [87]. BAIBA treatment was found to induce thermogenic genes in primary white adipocytes and human iPSC-derived white adipocytes. Notably, BAIBA only affected white adipocytes and had no effect on iPSCs differentiated into brown adipocytes. BAIBA administration to mice increased energy expenditure and had anti-obesity and anti-diabetic effects. The effects of BAIBA were blunted by a PPARα antagonist, suggesting that BAIBA works via a PPARα-dependent mechanism. 

### 3.7. Lactate and β-Hydroxybutyrate

Carrière et al. have reported that the metabolites lactate and β-hydroxybutyrate (BHB) induce beiging of WAT [88]. Based on data from a prior study, which revealed that levels of the monocarboxylate transporter MCT1, which is involved in the import of lactate, were strongly increased in BAT following endurance exercise [89], the authors tested the ability of cold to induce the transporter, and found that, indeed, cold acclimation also led to an increase in MCT1 levels that paralleled an increase in UCP1. Thus, they tested the ability of lactate to induce beiging in vivo. Curiously, they found that lactate treatment alone did not change UCP1 expression in inguinal or interscapular BAT. However, co-administration of lactate with the PPARγ ligand rosiglitizone led to an increase in thermogenic markers. This effect was MCT1 dependent, as knockdown or pharmacologic blockade of the transporter abrogated the effects of lactate. The authors also tested the effects of another monocarboxylate and MCT1 substrate, β-hydroxybutyrate, finding that it could also induce UCP1 in inguinal WAT. Thus, the metabolites lactate and β-hydroxybutyrate appear to act as beiging agents that may serve as a link between the metabolic stress of exercise and cold to the activation of UCP1-mediated thermogenesis. 

### 3.8. The Role of the Thyroid Hormone System in Beige Fat Thermogenesis

In humans, hypothyroidism leads to a decreased metabolic rate, increases in fat mass and intolerance to cold, while thyroid hormone excess produces opposing effects, eliciting metabolic increase, weight loss, and heat intolerance. Thyroid hormone has long been known to be involved in the regulation of metabolism as well as thermogenesis. However, despite the fact that this area has been studied for decades, the role of the thyroid hormone system in mediating ‘thyroid thermogenesis’ has proven elusive. The general thinking has been that thyroid hormone increases obligatory thermogenesis by accelerating numerous metabolic processes in multiple tissues, notably muscle [90,91]. Yet, in addition to obligatory thermogenesis, which accompanies all metabolic processes, the contribution of UCP1 and brown/beige fat to the thermogenesis elicited by TR activation is not yet clear. 

However, thyroid hormone signaling is clearly implicated in UCP1-dependent thermogenesis. The thyroid hormone system exhibits crosstalk with β-adrenergic signaling, the key regulator of UCP1 induction and activation and prior reports have suggested isoform-specific actions of the TRs in regulating adaptive thermogenesis, with TRβ being involved in the induction of UCP1, while the actions of TRα potentiate β-adrenergic signaling [92,93]. Likely due to the renewed interest in brown and beige fat, the role of thyroid hormone in mediating adaptive thermogenesis in adipocytes has begun to be revisited. In 2010, Lopez et al. reported that T_3_ administration reduces the activity of **5′** adenosine monophosphate-activated protein kinase (AMPK) in the ventromedial hypothalamus (VMH), leading to increased SNS output, activating BAT and UCP1-mediated thermogenesis. The authors demonstrated that ICV administration of T_3_, at low doses that resulted in no systemic effects when administered peripherally, increased BAT thermogenesis in a β_3_-AR dependent fashion [94]. A follow-up study by the same group found similarly that ICV-administered T_3_ also resulted in WAT browning, as expected from the mechanism of increased SNS output [95]. Another report, that relied upon central administration of T_3_, came to the conclusion that “UCP1 is essential for mediation of the central effects of thyroid hormones on energy balance”, showing that ICV administration resulted in increased UCP1-mediated thermogenesis [96].

Activation of the thyroid hormone receptors (TRs) has been shown to have numerous effects on metabolism that generally serve to oppose obesity and metabolic syndrome. While thyroid hormone administration has been used for weight loss, this is not a sanctioned use of the hormone, as thyroid hormone excess can have deleterious cardiovascular effects. To this end, numerous thyroid hormone analogues have been developed with the aim of safely harnessing the beneficial effects of TR activation, principally, the lipid lowering effects, while avoiding the untoward effects of thyroid hormone on the heart and bone. However, in addition to lipid lowering, many of these ‘thyromimetics’ also have actions on energy metabolism. Several years ago, my group reported that administration of the thyromimetic GC-1 could elicit a profound beiging of WAT in genetic and diet-induced models of obesity [97]. The observed beiging coincided with dramatic weight loss and anti-diabetic effects. Interestingly, all indicators of BAT activity decreased in ob/ob mice following GC-1 administration. That is, classical BAT became deactivated while beige fat activity was increased, potentially providing a unique model by which to study the effects of WAT beiging in the absence of effects on BAT. In this background, cold tolerance was found to increase dramatically in treated ob/ob mice, which are normally intolerant to temperatures as low as 4 °C. While we could not rule out that the increase in cold tolerance could be attributed, at least in part, to a decrease in heat loss, as TR signaling has been shown to affect vasoconstriction/dilation [98], GC-1 administration increases the metabolic rate at all temperatures tested between 4 and 30 °C, an effect consistent with increased heat production or thermogenesis that cannot be explained by decreased thermal conductivity, which would be expected to decrease the heat production required to maintain body temperature and, thus, lower the metabolic rate.

Two recent reports have investigated the basis for thyroid thermogenesis, the increased metabolic rate observed with thyroid hormone administration, both reporting this action to be UCP1-independent [99,100]. Dittner et al. administered T_4_ to WT or UCP1KO mice under thermoneutral conditions, which led to a doubling of the metabolic rate, and demonstrated that the metabolic elevation was equivalent in both groups [99]. Thus, the metabolic increase observed cannot be attributed to UCP1-dependent thermogenesis in brown or beige fat. They did find, however, that chronic T_4_ treatment produced a substantial increase in BAT UCP1 levels, which led to a very large UCP1-dependent metabolic increase in response to NE treatment. In a similar study from Johann et al., they reported that T_3_ administration raised the metabolic rate, increased body temperature, resulted in weight loss, and improved glycemic control [100]. Like the aforementioned study, they also found that T_3_ elicited a similar response in UCP1KO mice, indicating that these effects cannot be the result of UCP1-dependent thermogenesis. While T_3_ administration did result in WAT beiging, based on the induction of UCP1 and thermogenic genes, they showed further that glucose and lipid uptake is either reduced or unchanged in BAT and WAT following treatment and thus not contributing to the increased metabolism, consistent with the UCP1KO results. 

The last two studies shed light on the complex relationship between the thyroid hormone system and thermogenesis. They suggest that while thyroid administration increases the ‘potential’ for UCP1-mediated thermogenesis in beige or brown adipocytes, UCP1-dependent thermogenesis is not activated, as it is unnecessary to maintain body temperature due to increased obligatory thermogenesis emanating from UCP1-independent processes. The basis for this UCP1-independent thermogenesis is not clear, but it may be produced by muscle [100] or arise simply as a consequence of the increase in defended body temperature that is seen with thyroid hormone administration [99]. Unlike these investigations, using very similar methodologies, my group reported that GC-1 administration not only induced the genetic program of browning, but that beige fat was thermogenicity active at ambient temperature [97]. It is tempting to speculate that isoform and tissue-selective actions of the thyromimetic GC-1 may be responsible for the observed differences. Being TRβ selective, GC-1 would not be expected to exert the thermogenic changes in muscle observed by Johann and colleagues [100]. Without this obligate thermogenesis, SNS activation of brown and beige fat would again be expected, due to the thermal stress of ambient temperature for mice. Collectively, these studies indicate that TR activation can have both central and peripheral effects that are both UCP1-dependent and UCP1-independent and continued studies will be needed to discern whether these actions can be harnessed pharmacologically.

### 3.9. Non-UCP1 Mediated Non-Shivering Thermogenesis

The increasing prevalence of metabolic disease has spurred interest in discovering new therapeutic avenues for the treatment of obesity and its complications. In this light, there continue to be extensive efforts put forth toward the search for new effectors of thermogenesis. Specifically, many are working on identifying new mediators of non-UCP1-dependent non-shivering thermogenesis. Numerous factors that affect beige adipocyte thermogenesis have been reported recently, suggesting that these factors work via novel non-UCP1-mediated mechanisms [101,102,103]. However, UCP1 independence has not been strongly validated for most of these claims and nearly all reported beiging factors still require β-AR signaling for their metabolic effects. Thus, it seems likely that they may still function by either altering or potentiating SNS induction and activation of UCP1. These reports, as well as the ongoing search for such factors, prompt a fundamental question, *Does UCP1-independent non-shivering thermogenesis exist?* The existing evidence suggests that the answer may be both yes and no, depending on one’s specific definition, and where one is looking. β-AR agonists increase the respiration rate even in cells lacking UCP1 and β-adrenergic stimulation or cold exposure of UCP1KO mice has been reported to elicit a rise of the metabolic rate in some studies. Thus, it can likely be stated that based on experiments in rodents, some component of thermogenesis is non-UCP1 mediated. However, it is not at all clear whether there is one or more discrete mechanisms of non-UCP1-mediated thermogenesis that makes up this increased metabolic rate or whether the UCP1-independent thermogenesis sometimes observed in UCP1KO mice is mediated by the collective increased rate of numerous cellular processes, each individually contributing slightly to the total increase. If the latter scenario is the case, then it is likely that no individual non-UCP1-mediated action may mediate an effect that is likely to be pharmacologically targetable. 

It is my contention that the existence of UCP1-independent mechanisms of non-shivering thermogenesis of high magnitude is unlikely, particularly in adipocytes. First, evolutionary pressure would seem to have generally worked in the opposite direction, to ensure that metabolic processes are as efficient as possible in order to decrease the demand for energy and to build robust systems that *oppose* thermogenesis, which can be defined as the loss of energy as heat. Further, thermogenesis must be tightly regulated, as uncontrolled thermogenesis will result in the depletion of cellular energy and evoke cell death, which has been seen in cases of UCP1 overexpression [104], where UCP1 is constitutively active, and in the case of overdose of the chemical uncoupler and former diet drug dinitrophenol [4]. Thus, the evolutionary rationale for adipocytes to possess numerous backup systems to the tightly regulated and highly conserved system of UCP1-mediated thermogenesis is questionable. Secondly, the ability of UCP1KO mice to become acclimated to cold is commonly used as an argument for the existence of alternative non-UCP1 mechanisms of thermogenesis. However, unlike wild type mice, which cease to shiver once there is a sufficient abundance and activation of UCP1 to maintain defended body temperature, cold acclimated UCP1KO mice never stop shivering and die prematurely [105]. This suggests that there is no alternative mechanism that can compensate for the loss of UCP1 other than to increase the ability to maintain shivering thermogenesis. The prior point is supported by studies from pigs, which most agree do not express functional UCP1 protein [106]. Pigs are known to have poor thermal regulation [107] and have been reported to use shivering as their primary mechanism of thermogenesis [108]. Lastly, while any futile cycle will indeed evoke thermogenesis, this thermogenesis remains a ‘coupled’ process. That is, it still relies upon respiration to produce ATP, which is then consumed in futile cycling, the byproduct being heat. For this reason, investigations into alternative mechanisms of thermogenesis have typically revolved around muscle. Given the highly metabolic nature of muscle, its capacity for high levels of both ATP production and consumption, this is sensible and provides an evolutionary rationale for why the two most well established non-UCP1-mediated mechanisms of thermogenesis, shivering and malignant hyperthermia, take place there. Conversely, adipocytes, especially white adipocytes, are not highly metabolic, do not produce nor consume high levels of ATP, contribute only slightly to the overall metabolic rate, and would seem to be an unusual place to search for any coupled process of futile cycling. It has been well described elsewhere [109] that adipocytes do not have a high capacity to produce ATP, as they possess low levels of the ATP synthase complex [110,111]. Thus, the magnitude of any coupled thermogenic process will be limited by the capacity of adipocytes to produce ATP, making them incapable of producing effects of sufficient impact to mimic the actions of UCP1 and likely of any therapeutic relevance.

## 4. Conclusions 

The current understanding of beige adipocytes is in a stage of relative infancy. One major limiting factor is a lack of robust in vitro systems by which to study the mechanisms of beige fat development and thermogenic stimulation. An increased knowledge of the developmental origins of beige cells is likely a necessary step towards developing such cell-based systems. As nearly all beiging factors also affect classical brown adipocytes, the potential for beige adipocytes to alter systemic metabolism and elicit therapeutic effects still needs to be resolved. Without robust cell-based systems, it is also difficult to study how any discoveries in rodents may or may not translate to humans. However, given the high level of interest in beige adipocyte function and thermogenesis, it seems likely that future studies will start to unearth answers to the potential for beige adipocytes to affect human metabolic disease.

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
