# Peer review of "Beige Fat, Adaptive Thermogenesis, and Its Regulation by Exercise and Thyroid Hormone"

_biology, 2019, doi:10.3390/biology8030057_

Round 1

Reviewer 1 Report

The authors provide a review of current literature on regulation of beige adipogenesis. Papers are on beige adipogenesis are being published rapidly, and a review article summarizing the findings is necessary. This review represents an excellent synthesis of the current knowledge with citation of original discoveries, exceptional presentation of conflicting findings, and a bold but well supported perspective in the section on non-ucp1 mediated non-shivering thermogenesis. I recommend this review for publication with the minor revision I have recommended below. 

Minor revisions: 

1) In the section on beige adipose tissue I would recommend adding the cellular origin of beige adipocytes. What makes them distinct from brown adipocytes is that they arise from the same Myf5- precursor pool as white adipocytes, while brown adipocytes come from Myf5+ precursors. My guess is the omission of this is due to the complexity of the field where some studies show this distinction is not as simple as previously described (https://www.ncbi.nlm.nih.gov/pubmed/24942009?dopt=Abstract). However, this remains one of the major distinctions spoken of in the field and should be addressed here. Perhaps the addition of a sentence that says "One major distinction of beige adipocytes has been that they arise from the same Myf5- precursors as white adipocytes. More recently it has been observed that there is heterogeneity in their origin dependent on adipose depot with beige adipocytes from the anterior-subcutaneous, retroperitoneal, and posterior-subcutaneous WAT arising from Myf5+ precursors."

Author Response

Thanks for the constructive comment. I agree that the Myf5 status of beige adipocytes, being perhaps the only well accepted discriminating factor from brown adipocytes, should be addressed and appreciate the well constructed sentence. It has been added essentially verbatim. 

Best,

Kevin

Reviewer 2 Report

Phillips provides a timely and up to date review article on the topic of beige fat and its regulation by exercise and thyroid hormone. The article is very well written and I have only minor comments:

line 38/45: the danger of hyperthermia has also lead to the hypothesis that it may be more beneficial to trigger thermogenesis through the induction of heat loss by e.g. vasodilation (Warner et al. 2015 Adipocyte).

line 68: “receptor”

line 96: not all mammals possess functional BAT and vice versa there is also BAT in non-mammals, this should be phrased more carefully

line 104/106 etc: please check the nomenclature, if UCP1 protein is meant, it should be all capital

line 138/139: I would suggest to differentiate here between BAT for weight loss (which is unlikely to work, given the low amount of calories that can be burnt) and the role of BAT for type II diabetes to clear glucose (which may be working better).

line 167: “(refs)”

line 230: This paragraph should be phrased more critically in my opinion or even entirely eliminated, given the questionable validity of the Boström et al. publication (also in the light of the first author who was forced to retract several other publications for misconduct). At least it should be mentioned that the definitive proof of exercise induction of beige fat in the respective knock out animal models for irisin or its receptor is still missing, even several years after the original publication.

line 300: title should be italic and not bold

line 350: the study did look at browning, this was published as a follow up in 2017 (Martinez-Sanchez et al. JoE)

line 372: the heat loss regulation is mediated by TRa while GC1 acts mainly on TRb, so I would agree that this is not to be expected

line 400: I would also discuss here that this is not expected with a TRb agonist, as both muscle and brain are TRa tissues. This seems important to understand the phenotype of active beige fat upon GC1 administration (as here one would not expect muscle thermogenesis that can block BAT/beige fat activation).

line 429: There are several species that have lost UCP1 but can still survive in the cold (wild boars for example). I agree that in case of UCP1 presence, all other mechanisms are likely not of relevance for thermogenesis, but this should be worded more carefully, i.e. focussing on mechanisms for thermogenesis in adipose tissue vs mechanisms in muscle.

Author Response

Thank you to the reviewer for the numerous careful eyed corrections and suggestions. With two exceptions I will not respond point-by-point here as I agree with all the suggestions noted and the spirit of all points is now incorporated into the revised manuscript.

Regarding irisin, I have chosen to retain it, as it is still (sadly) part of the scientific literature. However, I have amended that section to note the reviewers points and to state that the consensus view is currently that irisin is a "myth". 

On my own perspective of the existence or non-existence of non-UCP1 non-shivering mechanisms being at play, I have tried to much more carefully craft that discussion to reflect that 1) muscle is the likely place for novel alternative thermogenic mechanism to exist (we know of two non-UCP1 mechanisms, shivering and malignant hyperthermia, which are included) and 2) that my major argument of the unlikelihood of non-UCP1 mechanisms is reserved for adipocytes. Reports from pigs are also now cited and discussed briefly. 

Thanks again for the insightful and helpful comments.

Best,

Kevin